# Cystic Fibrosis: A Descriptive Analysis of Deaths in a Two-Decade Period in Brazil According to Age, Race, and Sex

**DOI:** 10.3390/diagnostics13040763

**Published:** 2023-02-17

**Authors:** Luan Victor Frota de Azevedo, Fernanda Cristine Ribeiro Medeiros Cruz, Jéssica Paula Martins, Fernando Augusto Lima Marson

**Affiliations:** Laboratory of Cell and Molecular Tumor Biology and Bioactive Compounds, Laboratory of Human and Medical Genetics, Postgraduate Program in Health Science, São Francisco University, Bragança Paulista 12916-900, Brazil

**Keywords:** CFTR, diagnosis, distribution per race, epidemiology, genetic screening, human development index, neonatal screening, sweat test

## Abstract

The diagnosis of cystic fibrosis has improved in the last few years due to greater access to diagnostic tools and the evolution of molecular biology; the knowledge obtained has contributed to the understanding of its death profile. In this context, an epidemiological study was developed focusing on deaths from cystic fibrosis in Brazil from 1996 to 2019. The data were collected from the Data-SUS (Unified National Health System Information Technology Department from Brazil). The epidemiological analysis included patients’ age groups, racial groups, and sex. In our data, between 1996 and 2019, Σ3050 deaths were recorded, totaling a ≅330% increase in the number of deaths resulting from cystic fibrosis. This fact might be related to a better diagnosis of the disease, mainly in patients from racial groups that are not commonly associated with cystic fibrosis, such as Black individuals, Hispanic or Latino (mixed individuals/*Pardos*) individuals, and American Indians (Indigenous peoples from Brazil). Regarding of race, the Σ of deaths was: nine (0.3%) in the American Indian group, 12 (0.4%) in the Asian group, 99 (3.6%) in the Black or African American group, 787 (28.6%) in the Hispanic or Latino group, and 1843 (67.0%) in the White group. The White group showed the highest prevalence of deaths, and the increase in mortality was ≅150 times in this group, while, in the Hispanic or Latino group, it was ≅75 times. Regarding sex, the numbers and percentage of deaths of both male (*N* = 1492; 48.9%) and female (*N* = 1557; 51.1%) patients were seen to be relatively close. As for age groups, the >60-year-old group presented the most significant results, with an increase of ≅60 times in the registered deaths. In conclusion, in Brazil, despite the number of deaths from cystic fibrosis being prevalent in the White group, it increased in all racial groups (Hispanic or Latino, Black or African American, American Indian, or Asian individuals) and was associated with older age.

## 1. Introduction

Cystic Fibrosis (CF; OMIM: no 210700) is an autosomal recessive genetic disorder resulting from pathogenic variants in the *CFTR* (*Cystic Fibrosis Transmembrane Regulator*; at position 7q3.1) gene, which mainly affects White patients [1]. In Brazil, its estimated prevalence is 1:7500–15,000, varying according to the geographical region and the existing racial groups in each of these regions [2]. In such a context, in the last few years, the CF diagnosis, mainly in non-White patients, has improved, which seems to have resulted in better knowledge of the incidence of this disorder in Brazil and worldwide [3,4]. Worldwide, 162,428 people are estimated to be living with CF (data across 94 countries). Of these, it was estimated that 65% of patients are diagnosed, and 12% receive triple combination drugs [5]. At the same time, the epidemiological data have evidenced that CF is more common in Asian individuals (e.g., the prevalence of CF was around 1:128,434 in China, 3:100,000 in Bahrain, 1:15,876 in the United Arab Emirates, and from 1:43,321 to 1:100,323 in India; and the incidence of CF was around 1:2985 live births in Jordan, 1:5800 live births in Bahrain, and 1:350,000 live births in Japan), Hispanic or Latino individuals (e.g., the prevalence of CF was around 1:8500 in Mexico, from 1:6000 to 1:7000 in Argentina, ~1:1600 in Brazil (Euro-Brazilians), ~1:14,000 in Brazil (Afro-Brazilians), 1:4000 in Chile, and 1:3900 in Cuba), and African populations (e.g., average prevalence in Africa is 0.035 ± 0.03 using data from nine countries—most patients were imputed from Egypt and South Africa to estimate the average prevalence. Noticeably, outside of these countries, little data exist regarding CF in Africa, with no information available in 40 countries, mostly in Sub-Saharan Africa) than it was believed in the past [5,6,7,8,9,10,11,12,13]. However, it was impossible to determine the prevalence or incidence of CF among American Indians (Indigenous people).

In the CF diagnosis workflow, initial screening can be carried out with two Immunoreactive Trypsinogen doses in the newborn [first dose between the third and fifth day, and second dose up to 30 days of life]. In cases with altered Immunoreactive Trypsinogen, a complementary examination is required to confirm the diagnosis, including the measurement of chloride ions in the sweat test and the analysis of the *CFTR* pathogenic variants, mainly through *CFTR* gene sequencing [14,15,16,17,18]. The CF treatment involves different medicines and procedures, mainly physiotherapy to attenuate systemic clinical signals affecting the respiratory system [18]. The evolution in the disease management, mainly with the advent of precision medicine and personalized medicine [15], resulted in longer life expectancy for patients with CF, which, however, presents variability in different countries worldwide [4,19]. In this context, although there have been advances in CF diagnosis and treatment in the last few decades, morbidity and mortality still present high indices in patients with CF. Importantly, not all patients with CF have proper access to treatment [20], mainly regarding the new medications that modulate the expression of the CFTR protein [5]. For this reason and due to the diversity of *CFTR* variants associated with racial groups and Brazilian regions, the CF diagnosis and management in Brazil is still a challenge to be overcome, along with other diseases that affect the lungs and are caused by pathogenic variants in other genes (e.g., primary ciliary dyskinesia) [21,22].

Due to the evolution in its diagnosis and treatment, CF went through significant changes over the last few years, which might have resulted in a scenery of differentiated deaths. Therefore, this study aimed to evaluate CF deaths in Brazil from 1996 to 2019, according to the patients’ age, racial group, sex, and region of the country. Additionally, the total death indexes in 2017, 2018, and 2019 were correlated to the Brazilian Human Development Index (HDI), HDI—education (HDI-E), HDI—gross national income (HDI-GNI), and HDI—life expectancy (HDI-LE).

## 2. Materials and Methods

An observational retrospective study was carried out from 1996 to 2019 (a period with data available on the Brazilian Health Ministry online system) on CF in the E-84.9 category in the International Classification of Diseases (ICD) (10th version). The data were selected from the Unified National Health System Information Technology Department, Data-SUS—Brazil [23]. The death numbers were calculated considering the variables age (age groups: <4 years old; from five to 19 years old; from 20 to 59 years old; and >60 years old), racial group (self-family report and the individuals should identify only one category), sex (male and female), and location (place of residence) in the different regions of the country (North, South, Northeast, Southeast, and Midwest). We accessed and checked the Unified National Health System Information Technology Department, Data-SUS—Brazil for the last time on 10 December 2022.

The racial categories are according to the National Institutes of Health diversity programs and other reporting purposes as follows: (i) White; (ii) Black or African American; (iii) Hispanic or Latino (mixed race individuals/*Pardos*); (iv) American Indian (Indigenous peoples); and (v) Asian. The Brazilian Institute of Geography and Statistics (IBGE, *Instituto Brasileiro de Geografia e Estatística*) uses the same classification for races [24] and this classification was published before [25]. Although the meaning of race and ethnicity might overlap, the Merriam-Webster dictionary defines race as “a group sharing outward physical characteristics and some commonalities of culture and history,” whereas ethnicity is defined as “markers acquired from the group with which one shares cultural, traditional, and familial bonds” [26]. Thus, the term race was used and not ethnicity since it is most suitable for our variable.

Additionally, the information provided by the *Registro Brasileiro de Fibrose Cística*—REBRAFC (Cystic Fibrosis Brazilian Register) was collected for the 2009–2018 period, at http://portalgbefc.org.br/ingles/site/index.php, accessed on 10 December 2022, regarding the number of (i) new CF diagnosis per year, (ii) CF diagnosis from neonatal screening per year, and (iii) the number of deaths due to CF per year. We accessed the REBRAFC for the last time on 10 December 2022.

In addition, the HDI of the Brazilian states in 2017 was analyzed based on the data found in the Brazilian Human Development Atlas in 2021, developed in collaboration with the Applied Research Institute, the United Nations Development Program, and the João Pinheiro Foundation, accessed on 10 December 2022 using the link: http://www.atlasbrasil.org.br/ranking. The country’s HDI was evaluated considering the following factors: HDI—general index, HDI-E (dimension: knowledge; indicator: expected year of schooling and mean years of schooling), HD-GNI (dimension: a decent standard of living; indicator: GNI per capita), and HDI-LE (dimension: long and healthy life; indicator: life expectancy at birth).

The HDI (general, education, gross national income, and life expectancy) in 2017 was correlated to the percentage of deaths from CF per Brazilian federative unit, considering the death rate per 1000 live births in each of the Brazilian states in the 2017–2019 period and in the 1996–2019 period. The statistical analysis was carried out using Statistical Package for the Social Science software (IBM SPSS Statistics for Macintosh, Version 27.0). The Spearman correlation tests were applied to compare the HDI and the percentage of deaths from CF per Brazilian federative unit.

In the Spearman correlations, we considered the following cut-off points: (i) ±0.90–1.0, very high positive–negative correlation index; (ii) ±0.70–0.90, high positive–negative correlation index; (iii) ±0.50–0.70, moderate positive–negative correlation index; (iv) ±0.30–0.50 low positive–negative correlation index; and (v) 0.00–0.30, insignificant positive–negative correlation index. An alpha error of 0.05 was used in the statistical analysis. Our group used a similar statistical approach in a previous study about the association between the number of deaths of hospitalized patients due to severe acute respiratory syndrome in Brazil by the coronavirus disease and its association with HDI [27].

In the descriptive analysis, the absolute frequency and the relative frequency of the data collected in the Health System Information Technology Department, Data-SUS—Brazil (age groups, racial group, sex, and location (place of residence) in the different regions of the country) was shown, as well as the absolute and relative frequency of the data collected in the REBRAFC for deaths, absolute frequency only for new diagnoses, and mean, standard deviation, median, minimum value, and maximum value for age at death.

The graphic presentation was elaborated using the GraphPad Prism version 8.0.0 for Mac, GraphPad Software, San Diego, CA, USA, accessed on 10 December 2022 using the link: www.graphpad.com. XY graph plots were used to describe the relationship between the number of deaths according to age groups, racial groups, sex, and location (place of residence) in different regions of the country according to the timespan of data collection (per year). Additionally, a correlation matrix graph was used to present the correlations between the HDI, HDI-E, HDI-GNI, and HDI-LE with the death rate per 1000 live births in each Brazilian state in the 2017–2019 and 1996–2019 timespans.

The data used in our study are publicly available. By being anonymized, it is a consent-free study since it does not present risks to the research participants and was exempt from ethical approval by an Ethics Committee.

## 3. Results

Between 1996 and 2019, according to the DataSUS, the deaths caused by CF increased from 58 to 248 per year, totaling 3050 deaths and representing a ≅330% increase. This fact might be associated with a better diagnosis of this disease, mainly in patients of racial groups that are not commonly associated with CF, such as Black individuals, Hispanic or Latino individuals, and American Indians.

During the period of evaluation, the Brazilian Southeast (Σ1418; 46.5%) (the region with the largest number of inhabitants in the country) presented the highest number of deaths, with a fourfold increase during the study period (Figure 1A). Next, the Northeast (Σ597; 19.6%) and South (Σ604; 19.8%) regions presented a similar number of deaths when compared one to another, which was approximately twice as much as that observed in the North (Σ228; 7.5%) and Midwest (Σ203; 6.5%) regions of Brazil (Figure 1A).

The race variability observed with CF diagnosis is noticeable (Figure 1B). Among the racial components, the highest number of patients were White (Σ1843; 67.0%) and presented an increase in mortality of around 150 times. This fact was associated with the Caucasian origin of the disease and the Brazilian population foundation effect, which includes different components of migratory origin, mainly coming from migration events involving Portuguese, Italian, and Spanish groups. The second largest group included the Hispanic or Latino component (Σ787; 28.6%), which reveals the high degree of miscegenation in the Brazilian population, corresponding to approximately half of the death cases when compared to White individuals, representing an increase of around 75 times (Figure 1B). Finally, other racial groups such as Asian (Σ12; 0.4%) and Black or African American (Σ99; 3.6%) corresponded to a lower number of cases. However, the reports of CF cases in American Indian individuals (Σ9; 0.3%) also called attention since this group corresponds to an American race component usually dissociated from CF (Figure 1B). Another relevant finding is the reduction in the number of cases without information about race, which, even being self-declared, enables a better approach to the CF treatment, mainly regarding its phenotypical and genotypical variability.

When the deaths per age group were investigated, up to 2009, the percentage of deaths of infants below four years old was similar or over the other age groups evaluated. Initially, it corresponded to around 45% of total deaths (1996) and fell to approximately 15% in 2019. From 2009 onwards, as observed in Figure 2A,B, other age groups started to gain importance regarding the death profile of patients with CF. As for the age groups that showed a more significant increase in the number of deaths, the group >60 years old outstood with an increase of 60 times, followed by the group of patients aged between 20 and 59 years, with an increase of about eight times. Initially, the age group with patients >60 years old contributed to only 4% of the deaths recorded (1996), while, in 2019, this group contribution increased to over 40%.

Regarding sex, the numbers and percentage of deaths of both male (N = 1492; 48.9%) and female (N = 1557; 51.1%) patients were seen to be relatively close, without a significant discrepancy between these groups (Figure 3A,B).

The correlations between the number of deaths from CF in the 2017–2019 period and HDI, HDI-E, HDI-GNI, and HDI-LE and between the number of deaths from CF in the 1996–2019 period and HDI, HDI-E, HDI-GNI, and HDI-LE are presented in Figure 4. The number of deaths from CF from 2017 to 2019 shows a 40% correlation with HDI, 34% with HDI-E, 39% with HDI-GNI, and 39% with HDI-LE. These values represent a low correlation index. On the other hand, the deaths from CF in the 1996–2019 period showed a 66% correlation with HDI, 56% with HDI-E, 59% with HDI-GNI, and 65% with HDI-LE, which represent a moderate correlation index. Therefore, these results suggest that improvement of the country/region infrastructure, confirmed by the HDI increase, for example, might play a relevant role in the social impact caused by CF, which in this study is represented by the number of deaths caused by this disease in the study period.

We also described the number of deaths due to CF and new CF cases per year according to the CF Brazilian Register (Table 1). In the period evaluated (from 2009 to 2018), a continuous increase was observed in the number of CF cases reported and deaths caused by CF. However, the increase in deaths was neither proportional nor consistent with the data found in the National Register, as described by the Data-SUS. However, the information in the Data-SUS platform follows a suitable methodology to be used in this study, mainly regarding death classification, since there is no doubt that the reported patients died from CF rather than being patients with CF that died from other causes.

In the study period, an increased number of diagnoses through neonatal screening was observed due to the improved health policies promoted in Brazil, which resulted in an annual register of approximately 260 new cases. Finally, the data referring to the number of deaths from CF and HDI per federation unit were detailed (Table 2). These data were used to perform the correlation analyses between the previously described markers (Figure 4).

## 4. Discussion

Between 1996 and 2019, Σ3050 deaths were reported, representing a ≅330% increase in the number of deaths from CF in the five Brazilian regions. When the racial groups were investigated, deaths of American Indian (nine cases; 0.3%), Asian (12 cases; 0.4%), Black or African American (99 cases; 3.6%), Hispanic or Latino (787 cases; 28.6%), and predominantly White (1843 cases; 67.0%) individuals were recorded. The prevalence of deaths in the White group was confirmed with an increase of ≅150 times, followed by the Hispanic or Latino group with an increase of ≅75 times. As for the age groups, the >60-year-old group had the highest values, with an increase of ≅60 times. Regarding sex, the number of deaths was 1492 (48.9%) and 1557 (51.1%), respectively, for male and female patients.

In such a context, in the early 21st century, due to better knowledge of the disease, significant improvements were seen in the CF diagnosis methods, including the possibility of mapping the *CFTR* gene and predicting patients’ prognosis, since some patients, depending on the pathogenic variant, manifested a mild clinical condition. In contrast, others presented a severe clinical disease [2,19,28]. These factors, associated with neonatal screening and the evolution of precision medicine and personalized medicine, resulted in a promising development for the treatment of patients based on multidisciplinary care, enabling the inclusion of drugs such as Trikafta^®^ [Elexacaftor/Ivacaftor/Tezacaftor] (Vertex Pharmaceuticals™, Cambridge, MA, USA) [29,30,31,32] in the treatment workflow, which can modulate the pulmonary function evolution and reduce the pulmonary exacerbation and the chloride ion concentration in the sweat.

In Brazil, the infrastructure and resources of some regions might contribute to CF underdiagnosis and a worse prognosis for this disease [2]. In such context, Raskin estimated that only 10% of the patients with *CFTR* pathogenic variants were diagnosed in 2001 [7], including, for example, the lack of structure in some referral centers in São Paulo, which is considered one of the states with better conditions to carry out the sweat test [8,33].

In Latin America, for instance, CF diagnosis occurred at 3.7 years of age, and the mean age of death was 6.68 years in 1990 when only 10% of the patients reached adult age [34]. Additionally, in Brazil, despite neonatal screening being available all over the country, not all regions present the same level of availability of this examination, which compromises the CF diagnosis since screening is responsible for about 60% of the indication for the diagnosis to be confirmed. In addition, the disease treatment is not uniform, varying between Brazilian regions and different countries, which might contribute to the variability in the life expectancy, which is 20.5 years in South Africa and 49.7 years in Canada, for example [2,8,34]. Moreover, although the p.Phe508del (c.1521_1523delCTT; rs113993960—an inframe deletion variant) allele is the most frequent in Brazil, its contribution to the patient’s genotype is still controversial when studies and regions are compared, varying from 28% to approximately 50%. At the same time, the frequency of other *CFTR* gene variants is scarce [2,35,36]. Another difficulty in the diagnosis is identifying and understanding the phenotypical impact of *CFTR* variants categorized as rare or atypical [2,37]. Such variants might (i) present an uncertain meaning, (ii) be classified as large deletions or insertions (not evaluated in the gene panel sequencing), (iii) be found in intragenic regions, or (iv) be new variants that need to be better evaluated, including using computational predictors [36,38].

Most *CFTR* variants screened after the positive neonatal screening in the diagnosis workflow are associated with the White race group. In contrast, the pathogenic variants found in other racial groups, such as Black or African American and Hispanic or Latino, have not been well described yet. This fact hampers the diagnosis of non-Caucasian patients since a genetic examination of only some *CFTR* gene variants might produce a false negative result. The severity of this diagnosis deficit in different racial groups can also result in a worsened diagnosis of non-Caucasian patients since the later the identification of this disease occurs, the worse the quality of life and life expectancy of those patients are [7,8,39,40]. To confirm these situations, studies were carried out in different countries with *CFTR* gene sequencing to identify poorly elucidated variants and ensure the CF diagnosis is impartial. For example, Shum and co-workers performed the sequencing of an Indian child with a borderline result in the neonatal screening. They identified two variants, namely, p.Phe508del and c.870-1G>C (rs1351058559—a splice acceptor variant). This was the first time that the latter variant had been described in India [7]. Concomitantly, Mathew and co-workers reported the presence of the variant c.3623del (p.Gly1208Alafsx3; rs35396083—a frameshift variant) in an Arab child that had only been described once in the *CFTR*-2 base [10].

In Brazil, a study developed by da Silva Filho is the most significant to date in relation to the analysis of *CFTR* variants in patients with CF. The study included 3104 patients out of a population of 4654 individuals. Curiously, 236 *CFTR* gene variants were identified in that Brazilian study, out of which 114 were new genetic variants. After the *CFTR* sequencing, 2002 (64.5%) patients had positive results for the genotype test, 757 (24.4%) presented an inconclusive result, and 345 (11.1%) tested negative. An important factor reported in that study was that the proportion of individuals with a negative test result was higher in the North (45%) and Northeast (26%) Brazilian regions [2], and those regions are associated with non-White race groups.

The Brazilian racial diversity might contribute to the difficulties of diagnosis, which are associated with worse pulmonary function, nutritional condition, cognitive function, and early death. However, there was an improvement in the knowledge of the disease in all Brazilian regions. At the same time, the concentration of CF referral centers in Brazil should be re-evaluated since the South and Southeast regions present a greater concentration of centers due to the higher number of diagnoses [2,8,34]. However, other regions in the country also present a high number of patients, often associated with the disease’s nonclassical form. In those cases, the *CFTR* gene pathogenic variant that affects the race group related to the environment might explain the CF frequency and the death characteristic associated with it [2]. Another aspect to be taken into consideration is the role of the modifier genes since they might be related to race diversity, treatment response (including precision medicine and personalized medicine), and the response to environmental variations, including the risk of infection and colonization by different microbiological agents, which are mostly opportunistic [41,42,43]. Curiously, in Brazilian states where the Afro-descendent population predominates, such as Bahia and Maranhão, a high frequency of the 3120 + 1G>A (c.2988+1G>A; rs75096551—a splice donor variant) *CFTR* gene variant, which is associated with an African origin, was noticed. Therefore, the frequency of genetic variants varies in each state. This variability partly contributes to the mortality rate per region since these variants are also associated with disease severity and possible treatment by the precision and personalized medicine [44].

Regarding Brazilian miscegenation, *CFTR* gene variants originating in the Black or African American population were found with a different profile from that observed in the White patients. Macek and co-workers reported that African Americans have their own *CFTR* “common” subsets and that the most prevalent variant in their study was the 3120 + 1G>A variant, which was also found in Greek and Arab patients [45]. The origins of those variants are uncertain, and their high frequency in African Americans might be the outcome of a founder effect or a random derivation; in addition, the CF-causing variants tend to be specific to certain populations, varying according to the country of origin and racial group [34,45]. This situation justifies why this gene is found in regions such as North America, Greece, Saudi Arabia, and Latin America since these places were all included in slavery commerce in the past. Additionally, Black or African American patients with CF usually present less severe symptoms than those affecting White patients, possibly due to genetic factors that are not well known yet and the environmental modifying factors. These factors might be the reason why the mortality rates in this racial group are lower in Brazil. Therefore, in the clinical setting, CF must be included as a differential diagnosis, preventing the exclusion of this disease based on the patients’ race [34,45].

In such a context, part of the difficulty in diagnosing Afro-descendants occurs since CF was initially described as a disease of the White (Caucasian origin) population, turning this racial group into an exclusive target of studies on this disease for a long time. However, with the advancement of scientific discoveries, CF was observed in other racial groups. Therefore, some obstacles still exist involving the CF diagnosis in that racial group: the incidence of this disease is still lower in Black or African American patients, and it is considered relatively rare in this race, along with factors such as malnutrition, tuberculosis, and other diseases whose symptoms overlap those of the CF, causing its clinical diagnosis to be more difficult. In addition, as previously mentioned, there is still little information on the genetic variants that affect non-White races.

In a study by Stewart and co-workers investigating CF in members of the African diaspora, the main variants of Afro-descendants in the West were identified as follows: p.Phe508del (29.4% in America, Colombia, Brazil, and Venezuela); 3120 + 1G>A (8.4% in Brazil, America, and Colombia); p.Gly85Glu (c.254G>A; rs75961395—a missense variant) (3.8% in Brazil); 1811 + 1.6kbA>G (c.1680-886A>G; rs397508266—a splice donor variant) (3.7% in Colombia); and 1342-1G>C (c.1210-1G>C; rs397508178—a splice acceptor variant) (3.1% in America). Most of these variants were found in only one country, confirming the fact that they tend to be population specific. In this sense, the global data estimate that the number of CF-causing variants of European ascendence might be becoming closer to a plateau. On the other hand, there seem to be many more variants to be identified in non-White racial groups, such as those of individuals from the Black or African American racial group of the African diaspora [34]. Currently, in developed countries, CF patients’ life expectancy is 50 years old [46], while in Brazil, for example, confirming this scenery, the number of CF patients that live up to 60 years old or over has grown, which correlates with the number of deaths in this age group.

Moreover, regarding the increase in the number of deaths per region, some disparity is observed between the Southeast and other Brazilian regions. This might be explained by the larger number of patients with CF in that region since it concentrates the most populated states in the country, as well as to the historical process of European immigration to the South and Southeast regions. These regions also hold the highest HDI in the country, which generates positive impacts, mainly, regarding the disease diagnosis. The North region shows few cases since, despite being the largest region in the area, it is also the one with the lowest demographic density and whose origin presents a high contribution to the American Indian racial group [2]. The Northeast region is another area with few CF cases, which might be associated with the contribution of the racial component of African origin that forms that population and is related to several *CFTR* gene variants, which are not always easily diagnosed [2,35]. Despite that, in Brazil, there is still a predominance of European ancestry, which is confirmed by the high frequency of the pPhe508del allele found in Brazilian patients [2]. In Brazil and Africa, the CF incidence and prevalence cannot be estimated more precisely due to the different variants that remain undetected since several racial groups participated in the Brazilian population foundation. Moreover, the number of deaths of both male and female patients increased and kept similar values, with a slightly higher number in the female group, which agrees with the literature since the female sex is associated with worse survival rates, possibly due to the estrogen-harming effect. Thus, for example, women are more likely to develop exacerbations due to *Pseudomonas aeruginosa* [47].

Despite advancements, we believe that CF is still an underdiagnosed disease in Brazil, mainly in regions with lower HDI. The sample of diagnosed patients and the number of deaths might be different and higher than the findings of this study. In addition, the presence of some diseases that might mime a CF clinical condition, such as malnutrition, tuberculosis, and pulmonary infections commonly observed in African countries, hamper the identification of an etiological and sometimes clinical agent associated with CF [2,19,28,45].

The morbidity and mortality indices are possibly higher in patients with CF that were born and/or reside in regions where the neonatal screening, diagnosis, and/or management are incipient, depending on the disease genotype/phenotype relation, since the life quality and expectancy usually depend on an early diagnosis and aggressive treatment. Concomitantly, the importance of the implementation of the CF Brazilian Register becomes evident due to the actions developed by the Brazilian Group of Studies on CF, which resulted in countless advances in the diagnosis and treatment of this disease. This multidisciplinary group has fought for CF patients’ rights and currently has targeted the implementation of personalized and precision medicine for the management of the disease. The relevance of this group becomes even more evident when we consider the value associated with the use of precision and personalized medicine in Brazil [20]. Moreover, it is important to optimize the diagnosis of CF in Brazil and to provide full access to therapy to all patients to avoid the disparities in relation to race and other issues because it is well known that race can compromise the diagnosis and the treatment even when using new modulator strategies [40,48,49,50,51].

It is essential to highlight the influence of lifestyle and environmental factors on the outcomes of CF disease. For example, the patients should attend school as any other child does. However, they can face additional challenges related to feeding, treatments, and regular procedures (e.g., persistent infections may require time in the hospital). The onset of puberty and menstruation can be delayed by a few years. Children with CF tend to be smaller and thinner than other children, and the feeling of being different, based on body image stereotypes, can be stressful. It is also essential for caregivers to pay attention to patients who might neglect their treatments.

According to the CF News Today (accessed on 23 January 2023 at https://cysticfibrosisnewstoday.com/living-with-cystic-fibrosis/), the leading lifestyle recommendations from the National Institutes of Health include “not smoking and avoiding tobacco smoke, washing hands often to lower risk of infection, exercising regularly and drinking lots of fluids, and doing chest physical therapy. Even though patients may find it difficult to be physically active, it is shown that keeping active shown has benefits like increased exercise tolerance, respiratory muscle endurance, and sputum expectoration, reduced residual volume and rate of decline in pulmonary function, improvements in fluid balance and retention of serum electrolytes, and a lower risk of death. Nutrition can also drastically change patients’ digestive symptoms due to their deficiency in the pancreas and the accumulation of sticky mucus in the organ, which compromises the production of enzymes. The lack of enzymes hinders digestion and reduces the absorption of proteins and fats. In addition to taking enzyme, mineral, and vitamin supplements with every meal, patients are advised to eat regularly, prioritize food with more calories and proteins, and follow nutritional guidelines from a certified dietitian.”

The Healthline site, according to McDermott A. and medically reviewed by Alana Biggers A. (Updated on 3 August 2020; accessed on 23 January 2023 at https://www.healthline.com/health/living-with-cystic-fibrosis) described five tips for living well with CF that included: (a) Understand the treatment options (main aims: prevent lung infections and limit their severity, loosen and remove sticky mucus from the lungs, prevent and treat intestinal blockages, prevent dehydration, and have a proper nutrition). (b) Have a balanced diet (e.g., to support sodium loss and dehydration, as well as to increase the calorie intake due to the need for higher calories for patients with CF). (c) Create a workout plan (e.g., exercises to improve physical and emotional health). (d) Perform steps to avoid illness (e.g., wash hands after coughing or sneezing and after chest physical therapy; wash hands after petting animals, after using the bathroom, and before eating; wash hands after touching surfaces in public places; cover the mouth with a tissue when coughing or sneezing; throw the tissue away and wash hands; cough or sneeze into upper sleeve if a tissue is not available; do not cough or sneeze into hands; ensure all the vaccinations are current; receive an annual flu shot and stay at least six feet away from sick people and others with CF). (e) Connect with the community (e.g., depression can reduce the effectiveness of CF treatments and decrease lung function, so support groups can offer the opportunity to talk to other people who have experienced similar symptoms and experiences. Talk to the healthcare team or call your local hospital to see if there is a support group).

Finally, we hypothesized that three factors are the most important ones to decrease the number of officially reported deaths in Brazil: (a) increase the number of reference centers and the availability of sweat tests—including in regions with low population density and low incidence of White individuals per 1000 inhabitants; (b) implementation of neonatal screening for CF in neonatal screening tests in Brazil; and (c) higher accessibility of genetics tests to confirm the CF diagnoses mainly for those patients with the rarest pathogenic variants in the *CFTR* gene.

This study presents some limitations, such as the updated register of deaths being based on the data published by the Data-SUS and the REBRAFC. However, there is no consistency between this information and the two primary sources of information showing the different numbers of deaths. We opted for using the data provided by the Data-SUS due to the methodology employed in the result acquisition, according to the document titled “Sistema de Informações Sobre Mortalidade—SIM. Consolidação da base de dados de 2011” (Mortality Information System—SIM. Consolidation of the 2011 database). We did not obtain information regarding the *CFTR* genotype of patients that died of CF, which reduces the scope of the correlation between miscegenation and the number of deaths. In addition, we did not have access to other medical data of the patients, including comorbidity and medication to treat the clinical symptoms. Additionally, we did not consider the impact of precision and personalized therapy on the mortality rates because this modality of therapy is only recently being implemented in Brazil.

## 5. Conclusions

Although the prevalence of CF deaths in Brazil is higher in White patients, it in-creased in all racial groups (Hispanic or Latino, Black or African American, American Indian, or Asian individual) and has currently been associated with older age groups. During the study time, the diagnosis was improved, mainly due to sweat test availability and the broader inclusion of neonatal screening and genetic analysis, which, in general, enabled the diagnosis of mild phenotypes and caused non-White individuals to increase in the number of officially reported deaths due to CF in Brazil.

## Figures and Tables

**Figure 1 diagnostics-13-00763-f001:**
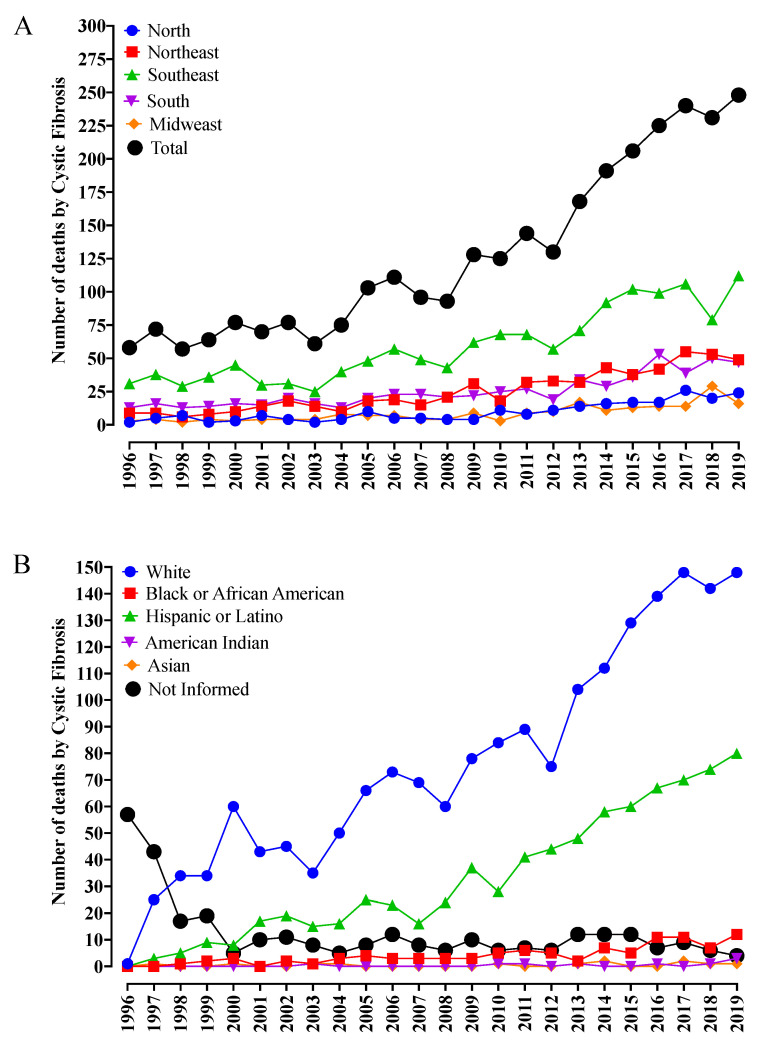
Number of deaths from cystic fibrosis (CF) in Brazil from 1996 to 2019 according to the Brazilian region (Figure 1A) and racial group (Figure 1B). (**A**) In 1996, the Southeast (the most populated Brazilian region) presented 31 reported deaths from CF. In 2019, the number increased from 31 to 112 cases, totaling 1418 deaths, with an increase of 3.6 times. The South region presented 13 deaths in 1996. This number rose to 47 in 2019, representing an increase of 3.6 times and totaling 604 deaths. The Northeast region had nine deaths in 1996, rising to 49 in 2019, demonstrating an increase of 5.4 times and 597 deaths. Finally, in 1996, the North region presented two deaths from CF, while the Midwest had three, which increased to 24 and 16, respectively, in 2019. This shows a 12-fold increase, totaling 228 deaths in the North region, while the Midwest region presented a 5.3-fold increase and 203 deaths. (**B**) Only one death from CF in the White group was recorded in 1996. In 2019, however, this number increased to 148. During this period, 1843 deaths were reported. No Hispanic or Latino, Black or African American, American Indian, or Asian individual died due to CF in 1996. On the other hand, in 2019, 80 deaths were reported in the Hispanic or Latino racial group, totaling 787 cases in the period analyzed; 12 in the Black or African American group, reaching 99 deaths; three in the American Indian group, totaling nine cases, and one in the Asian group, totaling 12 deaths. In addition to these cases, the Data-SUS death registers recorded deaths under the classification ‘not informed’. Within that group, 57 cases were reported in 1996 and only four cases in 2019, totaling 300 occurrences.

**Figure 2 diagnostics-13-00763-f002:**
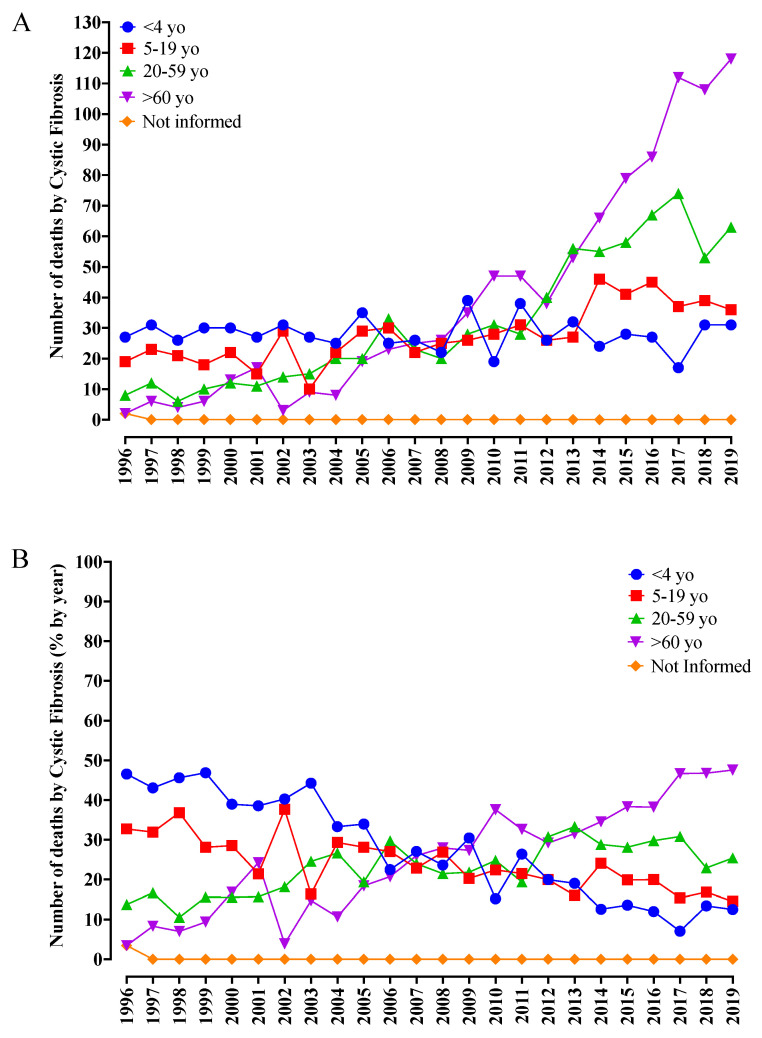
Number of deaths from cystic fibrosis (CF) in Brazil from 1996 to 2019 according to age group (Figure 2A) and relative percentage of deaths (Figure 2B). (**A**) The <4-year-old age group presented 27 cases in 1996 and 31 cases in 2019, resulting in 674 (22.1%) deaths from CF in the period analyzed. The age group from five to 19 years old presented 19 deaths in 1996 and 36 deaths in 2019, totaling 667 (21.9%) deaths from CF. The age group from 20 to 59 years old resulted in eight cases in 1996 and 63 deaths in 2019, resulting in 757 (24.8%) occurrences. Finally, the >60-year-old age group had two cases in 1996 and 118 cases in 2019, totaling 950 (31.2%) deaths from CF during the study period. (**B**) Considering all the age groups presented in the study, the >60-year-old age group showed steady growth over the years. Such a result suggests that CF management has contributed to longer life expectancy. yo, years old; %, percentage.

**Figure 3 diagnostics-13-00763-f003:**
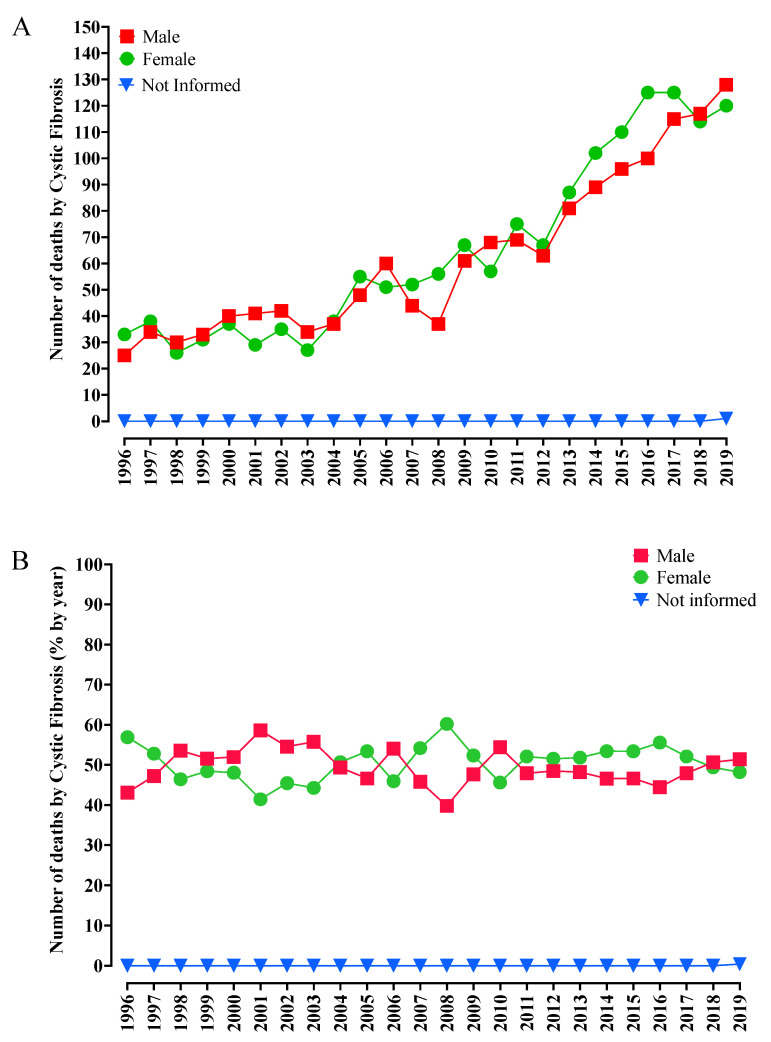
Mortality due to cystic fibrosis in Brazil from 1996 to 2019 according to sex (Figure 3A) and relative percentage of deaths (Figure 3B). (**A**) In 1996, 25 deaths of male patients and 33 deaths of female patients were reported, while, in 2019, the deaths reported involved 128 male patients and 120 female patients, totaling 1492 (48.9%) and 1557 (51.1%) deaths, respectively, in the study period. (**B**) The percentage of deaths in each sex group was similar, without significant discrepancy, in disagreement with the information found in the literature, in which the female sex is associated with higher mortality rates. %, percentage.

**Figure 4 diagnostics-13-00763-f004:**
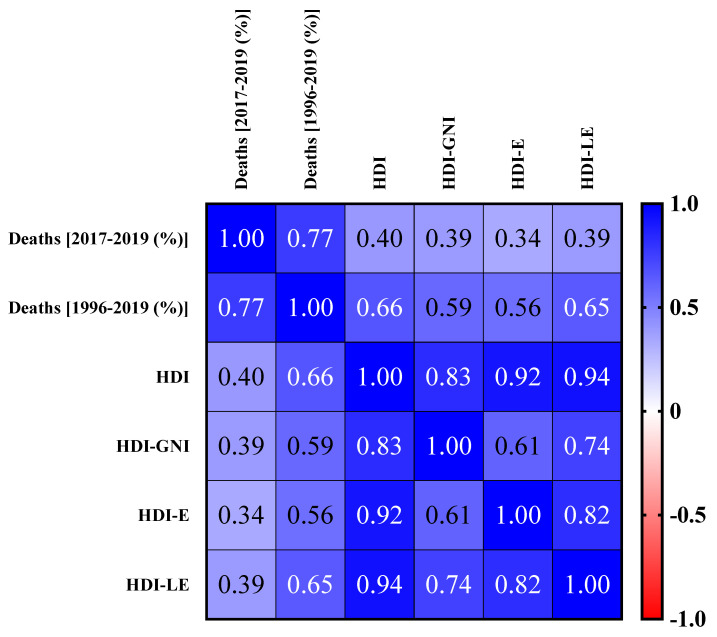
Correlation between the number of deaths from cystic fibrosis (CF) in the 2017–2019 period and the Human Development Index [HDI, HDI—education (HDI-E), HDI—gross national index (HDI-GNI), and HDI—life expectancy (HDI-LE)]; and between the number of deaths from CF in the 1996–2019 period and HDI, HDI-E, HDI-GNI, and HDI-LE. The Spearman correlations considered the following cut-off points: (i) ±0.90–1.0, very high positive–negative correlation index; (ii) ±0.70–0.90, high positive–negative correlation index; (iii) ±0.50–0.70, moderate positive–negative correlation index; (iv) ±0.30–0.50 low positive–negative correlation index; and (v) 0.00–0.30, insignificant positive–negative correlation index. An alpha error of 0.05 was used in the statistical analysis. %, percentage.

**Table 1 diagnostics-13-00763-t001:** Characterization of deaths and new diagnoses of cystic fibrosis in Brazil according to the Cystic Fibrosis Brazilian Register (*Registro Brasileiro de Fibrose Cística*—REBRAFC).

Year	Death		Total Patients	Age at Death (Years Old)			New Diagnoses	
	No	Yes		Mean ± SD	Median	Minimum–Maximum	Total	Neonatal Screening
2009	1032 (98.9%)	12 (1.1%)	1044	27.71 ± 23.09	21.44	6.25–80.00	244	79
2010	1432 (99.4%)	8 (0.6%)	1440	17.94 ± 12.03	19.81	1.33–37.05	318	112
2011	1550 (99.2%)	12 (0.8%)	1562	14.00 ± 6.64	13.40	1.88–27.30	257	112
2012	2107 (98.9%)	25 (1.2%)	2132	20.58 ± 11.14	21.07	2.42–54.93	270	131
2013	2208 (98.7%)	30 (1.3%)	2238	19.10 ± 11.31	20.64	0.41–41.73	295	153
2014	2525 (98.2%)	46 (1.8%)	2571	19.80 ± 10.15	18.98	0.27–41.64	277	173
2015	2905 (98.0%)	56 (2.0%)	2961	20.20 ± 10.57	18.38	0.29–43.78	259	135
2016	3154 (98.2%)	58 (1.8%)	3212	18.70 ± 14.80	14.60	0.60–76.59	245	148
2017	3328 (98.5%)	50 (1.5%)	3378	17.40 ± 9.26	15.70	0.40–40.60	276	170
2018	3289 (97.9%)	70 (2.1%)	3359	21.60 ± 16.20	18.40	0.25–40.60	286	166

SD, standard deviation; %, percentage.

**Table 2 diagnostics-13-00763-t002:** Number of death cases from cystic fibrosis and Human Development Index (HDI) per federative units in Brazil according to the Unified National Health System Information Technology Department, Data-SUS.

Federative Unit	2017–2019 (%)	1996–2019 (%)	HDI	HDI-E	HDI-GNI	HDI-LE
Acre	0.077	0.025	0.719	0.682	0.821	0.664
Alagoas	0.039	0.021	0.683	0.636	0.783	0.639
Amazonas	0.055	0.026	0.733	0.735	0.786	0.682
Amapá	0.039	0.018	0.740	0.710	0.820	0.695
Bahia	0.073	0.035	0.714	0.654	0.812	0.685
Ceará	0.072	0.031	0.735	0.717	0.818	0.676
Espírito Santo	0.114	0.051	0.772	0.732	0.850	0.740
Federal district	0.041	0.036	0.850	0.804	0.890	0.859
Goiás	0.129	0.041	0.769	0.740	0.822	0.747
Maranhão	0.068	0.021	0.687	0.682	0.764	0.623
Mato Grosso	0.057	0.034	0.774	0.758	0.825	0.742
Mato Grosso do Sul	0.068	0.028	0.766	0.710	0.847	0.748
Minas Gerais	0.096	0.050	0.787	0.753	0.875	0.741
Rio de Janeiro	0.072	0.050	0.796	0.763	0.858	0.769
Rio Grande do Norte	0.035	0.013	0.731	0.677	0.849	0.680
Rio Grande do Sul	0.129	0.074	0.787	0.729	0.849	0.787
Rondônia	0.048	0.020	0.725	0.703	0.776	0.699
Roraima	0.050	0.028	0.752	0.771	0.781	0.706
Santa Catarina	0.135	0.062	0.808	0.779	0.866	0.783
São Paulo	0.086	0.050	0.826	0.828	0.854	0.796
Sergipe	0.028	0.026	0.702	0.640	0.799	0.677
Pará	0.092	0.037	0.698	0.661	0.788	0.654
Paraíba	0.091	0.033	0.722	0.671	0.809	0.694
Paraná	0.089	0.053	0.792	0.764	0.843	0.771
Pernambuco	0.060	0.023	0.727	0.685	0.821	0.682
Piauí	0.045	0.030	0.697	0.666	0.771	0.660
Tocantins	0.095	0.029	0.743	0.727	0.811	0.696

E, education; GNI, gross domestic income; LE, life expectancy; %, percentage.

## Data Availability

Not applicable.

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
