# Peer review of "Cystic Fibrosis: A Descriptive Analysis of Deaths in a Two-Decade Period in Brazil According to Age, Race, and Sex"

_diagnostics, 2023, doi:10.3390/diagnostics13040763_

Round 1

Reviewer 1 Report

The English is too poor I can't follow the scientific story of the study also authors are not consistent with the universal scientific nomenclatures of the parameters used in the manuscript.

The study seems interesting and has an average value in this area of research and can be reconsidered after major English revision. 

Author Response

Reviewer 1.

The English is too poor I can't follow the scientific story of the study also authors are not consistent with the universal scientific nomenclatures of the parameters used in the manuscript.

The study seems interesting and has an average value in this area of research and can be reconsidered after major English revision.

Reply: The authors thank the reviewer for the important contribution. We sent the manuscript to be revised by a native English speaker. In addition, we tried our best to use universal scientific nomenclatures of the parameters, mainly, those nomenclatures used to describe the races of the patients.

We are attaching the English revision certificate.

Important: In the table, we cited the three parameters (age, race, and sex) using alphabetical order.

Reviewer 2 Report

In this article, Frota de Azevedo et al demonstrated that the number of deaths from cystic fibrosis was increased in all racial groups and was associated with older age. The article is very pleasant to read. The results are interesting and clear, while the conclusions are solid. I do have some minor comments/suggestions though: 

-It would be relevant to also pull out all other patient medical data including comorbidity and medication. This could have an impact on the interpretation of the results.

-In case these data are not available, I recommend that the authors acknowledge and discuss this as a limitation of the study.

Author Response

Reviewer 2.

In this article, Frota de Azevedo et al demonstrated that the number of deaths from cystic fibrosis was increased in all racial groups and was associated with older age. The article is very pleasant to read. The results are interesting and clear, while the conclusions are solid. I do have some minor comments/suggestions though:

-It would be relevant to also pull out all other patient medical data including comorbidity and medication. This could have an impact on the interpretation of the results.

-In case these data are not available, I recommend that the authors acknowledge and discuss this as a limitation of the study.

Reply: The authors thank the reviewer. We added the following excerpt in the text:

This study presents some limitations; namely, the updated register of deaths is based on data made available by the Data-SUS and the REBRAFC. However, there is no consistency between this information, and the two primary sources of information show different numbers of deaths. We opted for using the data provided by the Data-SUS due to the methodology employed in the result acquisition, according to the document titled “Sistema de Informações Sobre Mortalidade – SIM. Consolidação da base de dados de 2011” (Mortality Information System – SIM. Consolidation of the 2011 database). We did not obtain information regarding the CFTR genotype of patients that died of CF, which reduces the scope of the correlation between miscegenation and the number of deaths. In addition, we did not have access to other medical data of the patients, including comorbidity and medication to treat the clinical symptoms. Also, we did not consider the impact of precision and personalized therapy on the mortality rates because this modality of therapy is only being implemented recently in Brazil.

Important: In the table, we cited the three parameters (age, race, and sex) using alphabetical order.

Round 2

Reviewer 1 Report

 Journal of Diagnostics (ISSN 2075-4418)

Manuscript ID: diagnostics-2120901

Title 

Cystic Fibrosis: A Descriptive Analysis of Deaths in a Two-Decade Period in Brazil According to Age, Race, and Sex.

1.     I would like to thank the authors for the effort spent on the manuscript and for taking the reviewer's comments seriously.

2.     The English are significantly improved, and the scientific meaning of the study became clear.

ABSTRACT

1.     The abstract must be shorter and only the interesting results must be presented authors have missed a short conclusion to their abstract.

INTRODUCTION

1.     In this sentence of the manuscript, “the incidence of this disorder in our country and worldwide” authors must change “in our country” to in Brazil.

2.     In this part “At the same time, epidemiological data has evidenced that CF is more common in Asian, American Indian, Hispanic or Latino, and African populations than it was believed in the past” the frequency of CF must be added for each race.

MATERIALS AND METHODS

1.     If there is any Ethical approval and IRB registration number must be added 

2.     The last date of any webpage access must be added for example the access date of the Brazilian Health Ministry online system and Registro Brasileiro de Fibrose Cística – REBRAFC 96 (Cystic Fibrosis Brazilian Register)

3.     More details of the statistical model used to analyze the data must be added.

RESULTS

1.     Figures quality and colors can be improved  

2.     English is understandable and the results were well presented.

DISCUSSION

1.     Authors must avoid using OUR in the manuscript, information must be presented in their scientific meaning. 

2.     The authors are giving more importance to the molecular and genetic variation in their discussion; however, they are studying the CF deaths in Brazil from 1996 to 2019, according to the patient’s age, racial group, sex, and region of the country. It has been well established that the risk of CF and other diseases is the contribution of lifestyle, environmental factors, and genetic predispositions. Authors must give a major part of their discussion section to the lifestyle and environmental factors which is in the context of the study.

CONCLUSION

It must be improved we usually present in the conclusion section the most important funding of the study and the additional information that we get by doing this study these are not clear in this conclusion.

Author Response

Reviewer 1.

Manuscript ID: diagnostics-2120901

Title 

Cystic Fibrosis: A Descriptive Analysis of Deaths in a Two-Decade Period in Brazil According to Age, Race, and Sex.

  1. I would like to thank the authors for the effort spent on the manuscript and for taking the reviewer's comments seriously.

Reply: The authors thank the reviewer for contributing to improving our manuscript. In short, we included the following corrections:

  1. Abstract and text were edited according to the reviewer's comments.
  2. Images: we included the images using the maximum resolution output achieved from the GraphPad software.
  3. We included new references that we used to reply to the reviewer's comments.
  4. We included an excerpt to clarify the aims of discussing genetic tools and other diagnostic tests and their association with the improvement in the number of deaths in Brazil.

  1. The English are significantly improved, and the scientific meaning of the study became clear.

Reply: The authors thank the reviewer. In addition, the MDPi performs another final check before publishing the studies.

ABSTRACT

  1. The abstract must be shorter and only the interesting results must be presented authors have missed a short conclusion to their abstract.

Reply: Dear reviewer, we excluded some excerpts from the text, and we reviewed the conclusion section of the abstract.

INTRODUCTION

  1. In this sentence of the manuscript, “the incidence of this disorder in our country and worldwide” authors must change “in our country” to in Brazil.

Reply: We corrected the sentence of the manuscript.

  1. In this part “At the same time, epidemiological data has evidenced that CF is more common in Asian, American Indian, Hispanic or Latino, and African populations than it was believed in the past” the frequency of CF must be added for each race.

Reply: Dear reviewer, we added the corrections in the statement as recommended. We included the following excerpt:

Worldwide, 162,428 people are estimated to be living with CF (data across 94 countries). Of these, it was estimated that 65% of patients are diagnosed, and 12% receive triple combination drugs [5]. At the same time, epidemiological data has evidenced that CF is more common in Asian individuals (e.g., the prevalence of CF was around 1:128,434 in China, 3:100,000 in Bahrain, 1:15,876 in the United Arab Emirates, and 1:43,321 to 1:100,323 in India; and the incidence of CF was around 1:2,985 live births in Jordan, 1:5,800 live births in Bahrain, and 1:350,000 live births in Japan), Hispanic or Latino individuals (e.g., the prevalence of CF was around 1:8,500 in Mexico, 1:6,000 to 1:7,000 in Argentina, ~1:1,600 in Brazil (Euro-Brazilians), ~1:14,000 in Brazil (Afro-Brazilians), 1:4,000 in Chile, and 1:3,900 in Cuba), and African populations (e.g., average prevalence in Africa is 0.035±0.03 using data from nine countries – most patients were imputed from Egypt and South Africa to estimate the average prevalence. Noticeably, outside of these countries, little data exists regarding CF in Africa, with no information available in 40 countries, mostly in Sub-Saharan Africa) than it was believed in the past [5-13]. It was impossible to determine the prevalence or incidence of CF among American Indians (Indigenous people).

MATERIALS AND METHODS

  1. If there is any Ethical approval and IRB registration number must be added.

Reply: The authors thank the reviewer for the important comment. In our study, we used an open dataset from the Brazilian ministry of health and from the Cystic Fibrosis Brazilian Register. Both datasets are anonymized and can be consulted by the Scientific community and patients/caregivers. In this context, we included the following excerpt in the text:

The data used in our study were made publicly available. By being anonymized, it is a consent-free study since it does not present risks to the research participants and was exempt from ethical approval by an Ethics Committee.

  1. The last date of any webpage access must be added for example the access date of the Brazilian Health Ministry online system and Registro Brasileiro de Fibrose Cística – REBRAFC 96 (Cystic Fibrosis Brazilian Register).

Reply: The authors thank the reviewer. We included the information:

We accessed and checked the Unified National Health System Information Technology Department, Data-SUS – Brazil for the last time on December 10th, 2022.

We accessed the REBRAFC for the last time on December 10th, 2022.

  1. More details of the statistical model used to analyze the data must be added.

Reply: Dear reviewer, we corrected the material and methods section. In this context, we added several pieces of information about the study protocol, mainly, for the statistical model. We thank the reviewer for contributing to the improvement of this important section of the manuscript.

In addition, the HDI of Brazilian states in 2017 was analyzed based on the data found in the Brazilian Human Development Atlas in 2021, developed in collaboration with the Applied Research Institute, the United Nations Development Program, and the João Pinheiro Foundation, accessed using the link: http://www.atlasbrasil.org.br/ranking. The country’s HDI was evaluated considering the following factors: HDI – general index, HDI-E (dimension: knowledge; indicator: expected year of schooling and mean years of schooling), HD-GNI (dimension: a decent standard of living; indicator: GNI per capita), and HDI-LE (dimension: long and healthy life; indicator: life expectancy at birth).

The HDI (general, education, gross national income, and life expectancy) in 2017 was correlated to the percentage of deaths from CF per Brazilian federative unit, considering the death rate per 1,000 live births in each of the Brazilian states in the 2017-2019 period, and in the 1996-2019 period. The statistical analysis was carried out using Statistical Package for the Social Science software (IBM SPSS Statistics for Macintosh, Version 27.0). The Spearman correlation tests were applied to compare the HDI and the percentage of deaths from CF per Brazilian federative unit.

In the Spearman correlations we considered the following cut-off points: (i) ±0.90-1.0, very high positive-negative correlation index; (ii) ±0.70-0.90, high positive-negative correlation index; (iii) ±0.50-0.70, moderate positive-negative correlation index; (iv) ±0.30-0.50 low positive-negative correlation index; and (v) 0.00-0.30, insignificant positive-negative correlation index. An alpha error of 0.05 was used in the statistical analysis. Our group used a similar statistical approach in a previous study about the association between the number of deaths of hospitalized patients due to severe acute respiratory syndrome in Brazil by the coronavirus disease and its association with HDI [27].  

In the descriptive analysis, the absolute frequency and the relative frequency of the data collected in the Health System Information Technology Department, Data-SUS – Brazil (age groups, racial group, sex, and location (place of residence) in the different regions of the country) was shown, as well as the absolute and relative frequency of the data collected in the REBRAFC for deaths, absolute frequency only for new diagnoses, and mean, standard deviation, median, minimum value, and maximum value for age at death.

The graphic presentation was elaborated using the GraphPad Prism version 8.0.0 for Mac, GraphPad Software, San Diego, California USA, www.graphpad.com. XY graph plots were used to describe the relationship between the number of deaths according to age groups, racial groups, sex, and location (place of residence) in different regions of the country according to the timespan of data collection (per year). Also, a correlation matrix graph was used to present the correlation between the HDI, HDI-E, HDI-GNI, and HDI-LE with the death rate per 1,000 live births in each Brazilian state in the 2017-2019 and 1996-2019 timespan.

The data used in our study were made publicly available. By being anonymized, it is a consent-free study since it does not present risks to the research participants and was exempt from ethical approval by an Ethics Committee.

RESULTS

  1. Figures quality and colors can be improved.

Reply: Dear reviewer, in this manuscript version, we used the GraphPad with the highest possible output resolution. Also, we increased the size of all figures, and we printed the final version of the manuscript to visualize the image quality. In this approach, we noticed that the images, in this manuscript version, had a great quality to be published.

  1. English is understandable and the results were well presented.

Reply: The authors thank the reviewer, and we included minor corrections in the text.

DISCUSSION

  1. Authors must avoid using OUR in the manuscript, information must be presented in their scientific meaning. 

Reply: In the discussion section, we excluded the OUR from the text.

  1. The authors are giving more importance to the molecular and genetic variation in their discussion; however, they are studying the CF deaths in Brazil from 1996 to 2019, according to the patient’s age, racial group, sex, and region of the country. It has been well established that the risk of CF and other diseases is the contribution of lifestyle, environmental factors, and genetic predispositions. Authors must give a major part of their discussion section to the lifestyle and environmental factors which is in the context of the study.

Reply: The authors included the topics in the discussion section.

It is essential to highlight the influence of lifestyle and environmental factors on the outcomes of CF disease. For example, the patients should attend the school like any other child. However, they can face additional challenges related to feeding, treatments, and regular procedures (e.g., persistent infections may require time in the hospital). The onset of puberty and menstruation can be delayed by a few years. Children with CF tend to be smaller and thinner than other children, and the feeling of being different, based on body image stereotypes, can be stressful. It is also essential for caregivers to pay attention to patients who might neglect their treatments.

According to the CF News Today (accessed on January 23th, 2023 at https://cysticfibrosisnewstoday.com/living-with-cystic-fibrosis/), the leading lifestyle recommendations from the NIH include “not smoking and avoiding tobacco smoke, washing hands often to lower risk of infection, exercising regularly and drinking lots of fluids, and doing chest physical therapy. Even though patients may find it difficult to be physically active, it is shown that keeping active shown has benefits like increased exercise tolerance, respiratory muscle endurance, and sputum expectoration, reduced residual volume and rate of decline in pulmonary function, improvements in fluid balance and retention of serum electrolytes, and a lower risk of death. Nutrition can also drastically change patients’ digestive symptoms due to their deficiency in the pancreas and the accumulation of sticky mucus in the organ, which compromises the production of enzymes. The lack of enzymes hinders digestion and reduces the absorption of proteins and fats. In addition to taking enzyme, mineral, and vitamin supplements with every meal, patients are advised to eat regularly, prioritize food with more calories and proteins, and follow nutritional guidelines from a certified dietitian.”

The Healthline site, according to McDermott A. and medically reviewed by Alana Biggers A. (Updated on August 3rd, 2020; accessed on January 23th, 2023 at https://www.healthline.com/health/living-with-cystic-fibrosis) described five tips for living well with CF that included: a) Understand the treatment options (main aims: prevent lung infections and limit their severity, loosen and remove sticky mucus from the lungs, prevent and treat intestinal blockages, prevent dehydration, and have a proper nutrition. b) Have a balanced diet (e.g., to support sodium loss and dehydration, as well as, to increase the calorie intake due to the need for higher calories for patients with CF. c) Create a workout plan (e.g., exercises to improve physical and emotional health). d) Take steps to avoid illness (e.g., wash hands after coughing or sneezing and after chest physical therapy; wash hands after petting animals, after using the bathroom, and before eating; wash hands after touching surfaces in public places; cover the mouth with a tissue when coughing or sneezing; throw the tissue away and wash hands; cough or sneeze into upper sleeve if a tissue is not available; do not cough or sneeze into hands; make sure all your vaccinations are current; get an annual flu shot and stay at least six feet away from sick people and others with CF). e) Connect with the community (e.g., depression can reduce the effectiveness of CF treatments and decrease lung function, so support groups can offer the opportunity to talk to other people who have experienced similar symptoms and experiences. Talk to the healthcare team or call your local hospital to see if there’s a support group).

Finally, we hypothesized that three factors are the most important ones to decrease the number of officially reported deaths in Brazil: a) increase the number of reference centers and availability of sweat tests – including in regions with low population density and low incidence of White individuals per 1,000 inhabitants; b) implementation of neonatal screening for CF in neonatal screening tests in Brazil; and c) higher accessibility of genetics tests to confirm the CF diagnoses mainly for those patients with  the rarest pathogenic variants in the CFTR gene.”

CONCLUSION

It must be improved we usually present in the conclusion section the most important funding of the study and the additional information that we get by doing this study these are not clear in this conclusion.

Reply: We revised the conclusion as follows:

Although the prevalence of CF deaths in Brazil is higher in White patients, it increased in all racial groups (Hispanic or Latino, Black or African American, American Indian, or Asian individual) and has currently been associated with older age groups (>60-year-old group). During the study time, the diagnosis was improved, mainly due to the sweat test availability and broader inclusion of neonatal screening and genetic analysis, which, in general, enabled the diagnosis of mild phenotypes and non-White individuals the increase in the number of officially reported deaths due to CF in Brazil being evident.
